# VIDEO-TELLER: ENHANCING CROSS-MODAL GENERATION WITH FUSION AND DECOUPLING

## ABSTRACT

This paper proposes Video-Teller, a video-language foundation model that leverages multi-modal fusion and fine-grained modality alignment to significantly enhance the video-to-text generation task. Video-Teller boosts the training efficiency by utilizing frozen pretrained vision and language modules. It capitalizes on the robust linguistic capabilities of large language models, enabling the generation of both concise and elaborate video descriptions. To effectively integrate visual and auditory information, Video-Teller builds upon the image-based BLIP-2 model and introduces a *cascaded Q-Former* which fuses information across frames and ASR texts. To better guide video summarization, we introduce a fine-grained modality alignment objective, where the cascaded Q-Former's output embedding is trained to align with the caption/summary embedding created by a pretrained text auto-encoder. Experimental results demonstrate the efficacy of our proposed video-language foundation model in accurately comprehending videos and generating coherent and precise language descriptions. It is worth noting that the fine-grained alignment enhances the model's capabilities (4% improvement of CIDEr score on MSR-VTT) with only 13% extra parameters in training and zero additional cost in inference.

## 1 INTRODUCTION

Large language models (LLMs) have made significant advancements (OpenAI, 2023; Chowdhery et al., 2022; Bai et al., 2022), and have subsequently been extensively utilized in multimodal tasks such as image-to-text generation and video-to-text generation (Zhang et al., 2023a; Xu et al., 2023; Huang et al., 2023; Alayrac et al., 2022; Wang et al., 2022b), giving rise to a new class of models called multimodal large language model (MLLM). Prior to LLMs, video understanding models have been limited by the complexity of generated textual descriptions (Yan et al., 2023), and downstream video-to-text tasks are constrained to short-form generations such as single-sentence video captioning. With the incorporation of large language models, models such as Video-LLaMA, VideoChat and Video-ChatGPT (Zhang et al., 2023b; Li et al., 2023c; Maaz et al., 2023) are now capable of not only generating longer and nuanced video digests but also engaging in conversations grounded in video content.

To leverage the power of pretrained LLMs such as LLaMA 2 (Touvron et al., 2023) and Vicuna (Chiang et al., 2023) without incurring the forbidding cost of retraining LLMs, MLLMs such as BLIP-2 (Li et al., 2023a) have been proposed to integrate a trainable light-weight visual backbone with a frozen LLM via adaptor-like mechansims (such as the Q-Former proposed in BLIP-2). The expansion of BLIP-2 into the realm of video has quickly given rise to models such as Video-LLaMA (Zhang et al., 2023b), which incorporates both visual and auditory information by prompting frozen LLM with embeddings computing by two corresponding encoders. By relying on LLMs for modality integration, however, increases the computational cost during inference. Additionally, prior knowledge embedded in very large language models may negatively bias the generated video descriptions, leading to hallucination. Consequently, the issue of enhancing the accuracy of text generation in visual-language models while reducing the computational expense incurred has now become imminent.

In order to maintain the efficiency of model training while reducing computational overhead during inference, we propose cascaded Q-Former which fuses multi-frame visual information with auditory

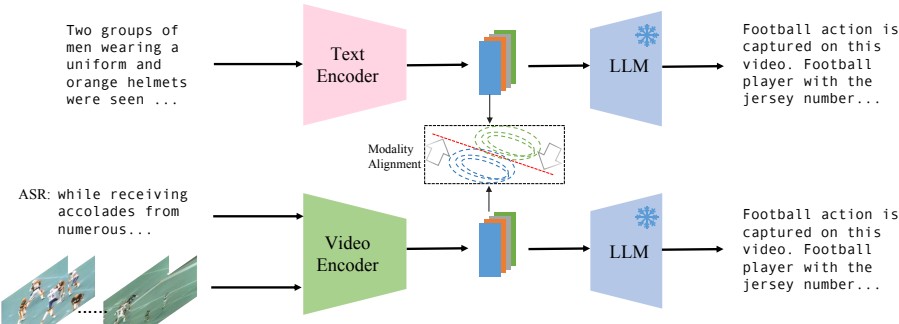

Figure 1: Overview of the proposed method. Here we show the detailed description generation (long-form text).

information prior to prompting LLMs, effectively reducing the computational overhead of LLM by half. Additionally, in contrast to Video-LLaMA's direct usage of raw audio, we leverage ASR information from videos as the representation for the audio modality to further enhance the model's comprehension capabilities. Due to the incorporation of additional modal information, to decouple crucial comments directly using methods similar to BLIP-2 becomes more difficult. Therefore, we propose the utilization of fine-grained modality alignment, thus enhancing the precision of content generated by the model and alleviating the issue of hallucination, we propose the utilization of fine-grained modality alignment as an auxiliary training approach. Figure 1 shows a high-level overview of the key concepts of Video-Teller.

We evaluated Video-Teller in both video-to-text generation and text-to-video retrieval tasks. Specifically, in video captioning, our approach achieves better results than the existing methods, such as HiTeA (Ye et al., 2022) with a smaller amount of data, thus confirming the effectiveness of fine-grained alignment and the integration of ASR. Additionally, in the task of long-text generation (video summarization), we obtained better performance (measured by BLEURT (Sellam et al., 2020)) than the baseline model with larger frozen LLMs such as Video-LLaMA and VideoChat (Zhang et al., 2023b; Li et al., 2023c).

Overall, the main contributions of this paper are as follow.

- We propose Video-Teller, a video-language foundation model that integrates both the visual and speech information. Video-Teller reduces the computational cost of modality fusion by incorporating visual and ASR information through a cascaded Q-Former before the LLM.

- We enhance video language learning by employing a text auto-encoder with LLM as the decoder to decouple textual features, enabling fine-grained alignment of video-text representations in an unsupervised manner. This approach improves the fusion of cross-modal information thus boosts the model's generation capability.

- In addition to providing demonstrations of Video-Teller output, we quantitatively compare our proposed method with two representative MLLMs, Video-LLaMA (Zhang et al., 2023b) and VideoChat (Li et al., 2023b).

## 2 RELATED WORK

The pursuit foundation models that integrate and understand multiple modalities like vision and text have received enormous impetus from the research community in recently years. Previously, foundation models were largely end-to-end trainable models with architectures like dual-encoders (Jia et al., 2021; Li* et al., 2022; Zeng et al., 2022; Bao et al., 2022), fusion-encoders (Li et al., 2019; Yang et al., 2022; Chen et al., 2020; Su et al., 2020; Li et al., 2020; Lu et al., 2019; Tan & Bansal, 2019), and encoder-decoders (Yu et al., 2022; Yan et al., 2023). However, these foundation models typically require fine-tuning of the entire model during adaptation, resulting in significant computational expenses. The advent of BLIP-2 (Li et al., 2023a) changed the multimodal landscape by introducing a lightweight adaptor module, the Q-Former, which utilizes learnable queries to facilitate

alignment of multiple modalities, reducing the need for fine-tuning the pre-trained language/visual models. Adaptor like modules have recently been remarkably successful in allowing multimodal researchers to tap into the tremendous natural language powers of large language models, with new models emerging every week such as InstructBLIP, VideoChat, Video-LLaMA and Mini-GPT4 (Dai et al., 2023; Li et al., 2023c; Zhang et al., 2023b; Zhu et al., 2023).

While adaptor modules like Q-Former can aid in merging input modalities, explicit alignments between modalities have traditionally been done via contrastive loss(He et al., 2020) between a single textual `[CLS]` token and the other modalities. Specifically, previous models either align `[CLS]` tokens of text features and visual features for sample-level modal alignment (Jia et al., 2021; Radford et al., 2021; Yu et al., 2022), or align `[CLS]` tokens with the rest of the tokens of the same modality using momentum (Yang et al., 2022). Such approaches enable modal alignment from a global perspective (via a single text token), but overlooks local information.

In contrast, few foundation models have explored fine-grained alignment between tokens across modalities during pre-training, which we argue is crucial for detailed understanding of complex input data such as videos. (Li et al., 2022) propose LOUPE, which learns fine-grained semantic alignment from the novel perspective of game-theoretic interactions. While (Shukor et al., 2022) leverage hierarchical cross-modal alignment loss for fine-grained modality alignment. These methods are highly effective, but they also tend to be more intricate. So in this paper, we propose to use a text auto-encoder to decouple the target text and use the decoupled feature to align with the video's hidden states.

## 3 METHOD

### 3.1 PRELIMINARIES: BLIP-2

BLIP-2 (Li et al., 2023a) is an image-text model designed to optimize training efficiency. It achieves this by utilizing pre-trained image encoders and frozen large language models. To address the modality gap, BLIP-2 introduces a lightweight Querying Transformer called Q-Former. Q-Former consists of learned queries and a transformer module with cross-attention. The input image initially undergoes the frozen Vision Transformer (ViT) to obtain an image patch token sequence. Subsequently, the learned queries interact with the image tokens through cross-attention. This process allows the input image to be encoded into a fixed-length sequence (as demonstrated in the paper, 32 tokens). These tokens are then projected and fed into the large language model to generate the corresponding text description of the image. Despite having $54\times$ fewer trainable parameters, BLIP-2 outperforms Flamingo80B (Alayrac et al., 2022). However, it should be noted that BLIP-2 is specifically designed to handle single-image inputs and cannot be directly applied to video-based applications.

### 3.2 MODEL ARCHITECTURE OF VIDEO-TELLER

As illustrated in Figure 2, Video-Teller is composed of two primary components. The first component is a video foundation model, which takes frames and ASR texts as input and incorporates a LLM as the language decoder. The second component is a text auto-encoder, which shares a similar structure with the video foundation model and also employs the same LLM as the language decoder. It is important to highlight that the text auto-encoder is exclusively utilized during the training phase and do not incur any additional computational cost during inference.

#### 3.2.1 VIDEO-TELLER

The extension of BLIP-2 to process video input can be tackled via multiple approaches. One approach is to encode each frame individually via image-based BLIP-2 model, and prompt LLM directly using the concatenated frame-level embeddings. This approach, although straightforward and powerful, incurs considerable computational overhead as the input sequence length of the frozen LLM is now multiple by the number of frames sampled. Additionally, this approach relies entirely on LLM to perform modality integration and alignment.

To address the aforementioned issue, we propose a cascaded Q-Former approach for integrating information from different video frames and texts generated by ASR. Let $\mathbf{V} \in \mathbb{R}^{F \times C \times H \times W}$ denote the input video frames, where $F$, $C$, $H$, and $W$ represent the number of frames, image channels,

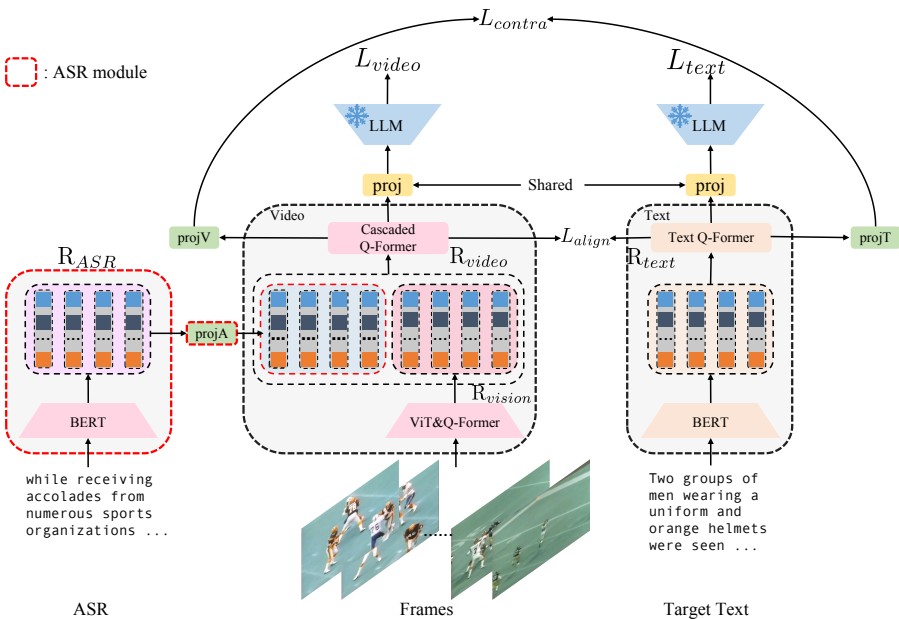

Figure 2: Overall architecture of the proposed model. The model consists of two primary branches. On the right-hand side, we have the text auto-encoder responsible for encoding the target text into a fixed-length representation denoted as $\mathbf{R}_{text}$. Conversely, on the left-hand side, we have the video module, which encodes the input video (comprising frames and ASR) into a video representation that shares the same shape as the text representation. Both of these representations are trained to reconstruct the target text by utilizing the LLM, while they are directly aligned through the Mean Squared Error (MSE) loss.

image height, and image weight, respectively. We utilize the vision encoder and Q-Former from BLIP-2 to individually extract the representation $\mathbf{R}_f \in \mathbb{R}^{Q_i \times E_i}$ for each frame, where $Q_i$ and $E_i$ indicate the number and size of query tokens in the image Q-Former. The aggregated visual features are obtained by concatenating all the image tokens from the Q-Former and are denoted as $\mathbf{R}vision \in \mathbb{R}^{FQ_i \times E_i}$.

For ASR text, we first use encoder-only BERT (Devlin et al., 2019) to process it and obtain the encoded text features. We use the last hidden states of the text features $\mathbf{R}_{ASR} \in \mathbb{R}_i^E$ as the ASR tokens to be combined with the visual features. In order for the combined ASR and visual features to be consumed by the LLM, we need to further downscale the dimension of the combined features since the total number of tokens is too large to be directly handled by the LLM. Towards this end, we employed another transformer to reduce the number of tokens and to fuse the information from the ASR and visual features. We name this added transformer cascaded Q-Former and it adopts the same BERT structure as the original Q-Former with fixed number of query tokens to produce a fixed length of result tokens. We concat ASR tokens $\mathbf{R}_{ASR}$ with visual tokens $\mathbf{R}_{vision}$ as input to the cascaded Q-Former. Finally, we gain the representation of the whole video $\mathbf{R}_{video} \in \mathbb{R}^{Q_v \times E_v}$, where $Q_v, E_v$ denotes the query number and embedded dimension of cascaded Q-Former. Here we manually split $\mathbf{R}_{video}$ into two components, where the first component includes the first token of $\mathbf{R}_{video}$ that is used for video-text contrastive learning, and the second component contains the remaining 32 tokens that is used for fine-grained modality alignment and video-grounded text reconstruction.

### 3.2.2 TEXT AUTO-ENCODER

We present a novel text auto-encoder that deviates from conventional models. Our approach leverages a frozen large language model as the decoder, while employing BERT and text Q-Former as the encoder to encode input text into a fixed-length representation ($32 \times 768$). Our objective is to find the mapping between the input text and the soft prompt (fixed-length representation) that enables the LLM to recover the same input text. Our experiments show that, for one-sentence captions, it

**Frames**: 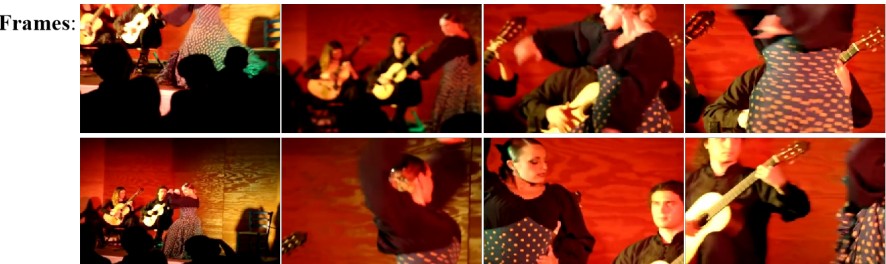

**ASR**：none.
**Video Captioning:**
Human annotated 1: a video of a dancer that dances to a guitar music.
Human annotated 2: a video of a woman dancing and performing.
.......
Human annotated 5: a flamingo dance performance with 2 guitars.
**Video Summarization:**
Human annotated 1: This video showed us a dance performance. The dancer was seen sitting in a chair at the beginning. As the music starts to play, she rised from the chair and dance to the rhythm. There were 2 guitars playing.
....
Human annotated 5: This is a video clip of a tango performance. In the video, a female dancer wearing a long skirt begins to dance. Next to him sat two guitar accompanists. The applause from the audience was very enthusiastic.

Figure 3: An example from Video-CSR. Here frames represent the video's vision information.

is sufficient to freeze the pretrained BERT module and only update weights of the text Q-Former. However, multi-sentence summaries cannot be reconstructed with frozen BERT encoder. As a result, in our subsequent experiments, the entire text encoder (including both BERT and text Q-Former) is trainable.

### 3.3 FINE-GRAINED MODALITY ALIGNMENT

As illustrated before, the text auto-encoder turns the input text into a fixed length intermediate representation, which covers the crucial information for text reconstruction. For video to text tasks, we aim at generating text with intermediate video representation. So we take the decoupled fixed length intermediate representation from text auto-encoder as intermediate target for video foundation model. This means we align video with corresponding text not only through the video-grounded text reconstruction but also the hidden feature's consistency, namely fine-grained modality alignment. Our proposed method is different from previous, instead of using game-theoretic (Li et al., 2022) or hierarchical cross-modal alignment loss (Shukor et al., 2022) as we directly utilize the encoded tokens in our text auto-encoder as a learning target.

## 4 EXPERIMENT

We evaluate our proposed method on several downstream tasks, including video captioning, video summarization and video retrieval.

### 4.1 SETUP

**Datasets**    We test our method's video understanding capability on MSR-VTT (Xu et al., 2016) and Video-CSR (Liu et al., 2023). MSR-VTT is a large-scale video benchmark for video understanding, especially generating video captions. It covers 10K videos and each was annotated with about 20 natural sentences. There are 6513 videos for training and 2990 videos for testing. It should be noted that as MSR-VTT doesn't provide ASR, we use "none." as the input to our ASR branch. Video-CSR is a newly released large-scale video benchmark for video understanding, covering roughly 5000 videos ranging from 15 seconds to 1 minute with each video annotated by human. There

Table 1: Results for video caption on MSR-VTT. w/o A means without fine-grained modality alignment. SCST means Self-Critical Sequence Training (Rennie et al., 2017). **Video-Teller achieves similar performance with its counterparts but uses much less PreTraining Data.**

| Model | #PT Data | B@4 | M | R | C |
|---|---|---|---|---|---|
| HiTeA (Ye et al., 2022) | 17M | 49.2 | 30.7 | 65.0 | 65.1 |
| VideoCoCa (Yan et al., 2023) | 3B | 53.8 | - | 68.0 | 73.2 |
| GIT (Wang et al., 2022a) | 0.8B | 53.8 | 32.9 | 67.7 | 73.9 |
| GIT2 (Wang et al., 2022a) | 12.9B | **54.8** | 32.9 | **68.2** | **75.9** |
| Video-Teller w/o A | 4.5M | 47.9 | 32.4 | 65.5 | 68.0 |
| Video-Teller | 4.5M | 49.2 | 33.0 | 66.4 | 72.0 |
| Video-Teller (SCST) | 4.5M | 49.4 | **33.4** | 67.0 | 74.5 |

are 5 captions and 5 summaries for each video. Summaries are long captions that includes more details about the subject and activities in the video. The average length of captions is 12.71 and the average length of summaries is 62.93 in Video-CSR. We adopt Video-CSR since its videos contain rich ASR information and is suitable to evaluate our framework with both visual and ASR input. Videos in this dataset can be divided into two parts, one part with rich ASR information while the other part with little ASR infromation. The ratio of videos with ample and limited ASR information is approximately 1 to 2. In cases that the video's ASR conveys only little information, we use the text "none." as the ASR input.

For experiments on MSR-VTT, we use the WebVid-2M (Bain et al., 2022) and CC3M (Sharma et al., 2018) (used as static video) for pre-training. While in experiments on Video-CSR, we collect a pre-training dataset consists of 100K YouTube videos, where each video has 5 captions and 5 summaries generated by GPT-3.5 with the videos' metadata from YouTube, which covers description, ASR, comments and so on. The generated captions contains the key information of the video, but may miss some essential visual information if it is not described by the video's metadata. An example of a video from Video-CSR is shown in Figure 3.

**Model Configurations** We construct our model directly from the pre-trained BLIP-2, leveraging its extensive prior knowledge of images. For text auto-encoder, we use $BERT_{base}$ to process the raw input. And then we use the first five layers of $BERT_{base}$ as the text Q-Former. 33 learnable queries are used to project the text into the fix-length representation with cross-attention where the first one is for video-text contrastive learning. For the cascaded Q-Former, we construct it with the first 5 layers of the pre-trained $BERT_{base}$ with 33 learnable query tokens. Empirically, we find that using more layers on the cascaded Q-Former and text Q-Former will deteriorate the performance. For ASR, the weights of $BERT$ in its encoder is shared with the text auto-encoder. Totally, there are about 307M trainable parameters containing the image Q-Former, video Q-Former, text encoder, text Q-Former and a few linear projection layers. We apply pretrained opt-6.7b (Zhang et al., 2022) as our frozen LLM. For each video input, we sampled 8 frames as the vision representation.

**Training and Evaluation** We adopt a three-stage training process for our model. For the first stage, we pretrain the text auto-encoder. We use 32 Tesla-V100 GPUs, with a batch size of 8 on each individual GPU, and conducted training for two epochs. For the second stage, we train the whole model on video captioning or video summarization using 64 Tesla-V100 GPUs, with a batch size of 8 for video captioning and 6 for video summarization on each individual GPU. We train the model for another 2 epochs. For the third stage, we further finetune the model with 32 Tesla-V100 GPUs for 3 epochs on MSR-VTT and 10 epochs on Video-CSR.

## 4.2 VIDEO CAPTIONING

We conducted experiment on two datasets, MSR-VTT and Video-CSR. As Video-CSR is a newly released dataset, we implement baselines with VideoCoCa (Yan et al., 2023) by adding a similar ASR fusion module to facilitate it to extract information from both frames and ASR text. We initialize VideoCoCa with CoCa pretrained on LAION-5B (Schuhmann et al., 2022). We name the VideoCoCa model with the added ASR fusion module VideoCoCa (ASR). Results on MSR-VTT

Table 2: Results for video caption on Video-CSR. w/o A means without fine-grained modality alignment. For Zero-Shot, both models are trained 100K videos from pretraining dataset.

| Model | ASR | Finetuned | | | | Zero-Shot | | | |
|---|---|---|---|---|---|---|---|---|---|
| | | B@4 | M | R | C | B@4 | M | R | C |
| VideoCoCa | No | 6.2 | 11.0 | 23.8 | 18.7 | 2.1 | 10.7 | 18.7 | 5.7 |
| VideoCoCa (ASR) | Yes | 7.1 | 11.9 | 25.0 | 22.1 | 2.8 | 11.4 | 19.7 | 9.1 |
| Video-Teller w/o A | Yes | 7.2 | 12.7 | 26.3 | 21.9 | 3.5 | 12.2 | 22.2 | 13.9 |
| Video-Teller | Yes | **10.4** | **14.7** | **28.7** | **30.7** | **5.6** | **14.2** | **24.0** | **19.9** |

Table 3: Results for video summarization on Video-CSR. w/o A means without fine-grained modality alignment.

| Model | #PT Data | Finetuned | | | Zero-Shot | | |
|---|---|---|---|---|---|---|---|
| | | BLEURT | R | C | BLEURT | R | C |
| VideoCoCa | 0.5M | 29.6 | 19.3 | 2.9 | 28.8 | 18.6 | 3.0 |
| VideoCoCa (ASR) | 0.5M | 36.8 | 22.4 | 9.5 | 31.0 | 20.1 | 8.1 |
| Video-LLaMA | - | - | - | - | 39.3 | 19.2 | 2.1 |
| VideoChat | - | - | - | - | 42.8 | 22.6 | 15.2 |
| Video-Teller w/o A | 0.5M | 45.2 | 22.4 | 9.7 | 41.2 | 20.1 | 7.1 |
| Video-Teller | 0.5M | **47.1** | **23.5** | **11.2** | **43.3** | 21.3 | 9.0 |

can be found in Table 1 and results on Video-CSR can be found in Table 2. All results are reported on BLEU-4 (B@4), METEOR (M), CIDEr (C) and ROUGE-L (R).

We also test the model applying self-critical sequence training, which is a REINFORCE algorithm that directly optimize the CIDEr metric (Rennie et al., 2017). Those results demonstrate Video-Teller's strong video description though using limited videos for pre-training compared with other models.

### 4.3 VIDEO SUMMARIZATION

We evaluate performance of video summarization on Video-CSR. This dataset covers 5000 videos. We randomly choose 1200 videos for testing, while the rest are used for fine-tuning the models. It is important to mention that the ratio of videos with ample and limited ASR information in the test set and training set is both approximately 1 to 2. We compare with four baseline models: VideoCoCa, VideoCoCa (ASR), Video-LLaMA (Zhang et al., 2023b), and VideoChat (Li et al., 2023b). Among them, VideoCoCa and Video-LLaMA only uses visual input and VideoCoCa (ASR) and VideoChat uses both visual and ASR input. We also evaluate both zero-shot and finetuned performance. For the metrics, we choose BLEURT (Sellam et al., 2020) as the main metrics. We also report results with CIDEr (C) and ROUGE-L (R). Results can be found in Table 3. After calculating the metrics, we randomly select 20 generated sentences from different models. We manually ranked each result to assess their level of consistency with various indicators and find that semantic-related evaluation metrics such as BLEURT (Sellam et al., 2020) are more suitable than metrics based on string matching for long text evaluation. The results also indicate that Video-Teller has achieved certain advantages in video summarization compared to other models.

### 4.4 ABLATION EXPERIMENTS

While our model exhibits commendable performance in the text generation task, we remain skeptical about the extent to which the inclusion of ASR and fine-grained alignment can genuinely enhance its performance. Consequently, we undertake ablation experiments and assess them using Video-CSR and MSR-VTT dataset. Results on Video-CSR can be found in Table 4. Results for MSR-VTT are in Table 4. As MSR-VTT doesn't provid ASR, we only test the influence of alignment and contrastive loss. From the results, we can observe that on Video-CSR our model's performance declines when either ASR or fine-grained alignment is absent. This demonstrates the effectiveness

Table 4: Results for video captioning. Here w/o A means without align while w/o C means without contrastive learning. We also use w/o ASR represents without ASR.

| Model | Finetuned | | | | Zero-Shot | | | |
|---|---|---|---|---|---|---|---|---|
| | B@4 | M | R | C | B@4 | M | R | C |
| Results on Video-CSR | | | | | | | | |
| Video-Teller w/o ASR | 4.7 | 9.8 | 22.8 | 13.1 | 2.2 | 10.5 | 19.7 | 13.4 |
| Video-Teller w/o A | 7.2 | 12.7 | 26.3 | 21.9 | 21.9 | 12.2 | 22.2 | 13.9 |
| Video-Teller w/o C | 10.3 | 14.7 | 28.5 | 30.4 | 5.6 | 14.1 | 24.0 | 19.8 |
| Video-Teller | **10.4** | **14.7** | **28.7** | **30.7** | **5.6** | **14.2** | **24.0** | **19.9** |
| Results on MSR-VTT | | | | | | | | |
| Video-Teller w/o A | 47.9 | 31.5 | 65.3 | 69.6 | 12.4 | 17.6 | 36.3 | 24.6 |
| Video-Teller w/o C | 48.4 | 32.9 | 65.7 | 70.9 | 13.4 | 18.7 | 38.5 | 25.4 |
| Video-Teller | **49.2** | **33.0** | **66.4** | **72.0** | **15.6** | **19.6** | **40.1** | **26.9** |

Table 5: Results for video retrieval on MSR-VTT, where w/o A means without fine-grained alignment.

| Model | #PT Data | Finetuned | | | Zero-Shot | | |
|---|---|---|---|---|---|---|---|
| | | R@1 | R@5 | R@10 | R@1 | R@5 | R@10 |
| Video-Teller w/o A | 4.5M | 33.1 | 57.8 | 65.9 | 41.0 | 67.1 | 77.3 |
| Video-Teller | 4.5M | 33.5 | 57.5 | 66.1 | 40.7 | 67.5 | 77.7 |

of our approach on real-world scenario datasets. Result on MSR-VTT captioning also shows the fine-grained alignment improves the performance.

## 4.5 VIDEO RETRIEVAL RESULTS

Though achieving strong results on video generation task with fine-grained modality alignment, it still needs to be verified whether the method will have an impact on the accuracy of retrieval. Through ablation experiments, it's demonstrated that fine-grained modality alignment enhances the cross-modal generation capability of the model without affecting its retrieval accuracy. Result can be found in Table 5. The model is pre-trained with WebVid-2M (Bain et al., 2022) and CC3M (Sharma et al., 2018). From above result, we have demonstrated that fine-grained alignment can enhance the generation capability of the model without affecting video retrieval task.

## 5 ANALYSIS

As shown before, we find that Video-Teller, with limited video data for pre-training, achieves strong performance both on video summarization and video captioning. We will analyze the improvements to the model that our proposed method brings in terms of hallucination of description.

Similar to LLM, Video-Teller is bothered by hallucination. it tends to fill in incorrect information, especially when generating detailed description. We evaluated the severity of different models' hallucination through manual assessment. Specifically, we randomly selected 50 generated results from the test set of Video-CSR and categorized them into three types: no hallucination, slightly hallucination, and severe hallucination, based on the comparison between the generated content and manually annotated content. The ratios for each model are illustrated in Table 7. We also provide criteria for rating different levels of hallucination in Table 6.

Table 6: Criteria of rating hallucination.

| Hallucination level | Description |
|---|---|
| no hallucination | The predicted summary delineates events that are entirely congruous with the actual video, albeit with potential omissions in its depiction. |
| moderate hallucination | The predicted summary portrays events that are largely congruent with the actual video, albeit with some minor deviations in certain details. |
| severe hallucination | The predicted summary depicts events that are starkly divergent from the actual video. |

Table 7: Hallucination ratio of different models. w/o A means trained without fine-grained alignment.

| Model | no hallucination | moderate hallucination | severe hallucination |
|---|---|---|---|
| VideoCoCa (ASR) | 0.60 | 0.26 | 0.14 |
| Video-LLaMA | 0.26 | 0.40 | 0.34 |
| Video-Teller w/o A | 0.40 | 0.26 | 0.34 |
| Video-Teller | 0.56 | 0.24 | 0.20 |

Based on the findings presented in Table 7, it is evident that models utilizing LLMs face a more pronounced issue of hallucination. This can be attributed to the limited information provided by the visual encoder, forcing the LLM to heavily rely on imaginative processes to complete the description. In contrast, VideoCoCa, which does not employ an LLM, exhibits a relatively milder form of hallucination. This difference can be explained by VideoCoCa's tendency to generate shorter descriptions when faced with insufficient information, thereby reducing the generation of extraneous content. Conversely, the extensive prior knowledge of the LLM engenders the production of erroneous information. With our fine-grained alignment, Video-Teller is able to significantly reduce the rate of hallucination, with the no hallucination rate increased from 0.40 to 0.56, and severe hallucination rate diminished from 0.34 to 0.20. This indicates that the fine-grained alignment enforces the encoded video tokens $\mathbf{R}_{video}$ to be more relevant to the semantics of the target caption/summary and thus reduces hallucination. We provide a demo which shows model trained without fine-grained alignment suffers more from hallucination in Figure 4 in the appendix.

As we could see in Figure 4, this video belongs to the category with a high ASR (Automatic Speech Recognition) content. Therefore, in order to generate its summary, it is necessary to make better use of the ASR information. From the ASR information, we can infer that this video discusses the relevant aspects of the decline in clean energy prices, just as predicted by Video-Teller. However, we can observe that without alignment, the model's description includes specific price changes that cannot be extracted from the video.

## 6 CONCLUSION

In this paper, we propose Video-Teller, a robust video-text foundation model that attains impressive performance on video-to-text tasks, encompassing both concise and comprehensive descriptions. Video-Teller leverages the rich speech information contained in the videos to enhance the model's understanding of the video. Simultaneously, it utilizes pre-trained visual models and large language models to reduce training costs while maintaining the impressive performance. Furthermore, we employ a standalone text auto-encoder to learn the proper intermediate language tokens that guides the learning of the video foundation model, which boosts the decoupling of the fused multi modality information. Extensive experimental results demonstrate the impressive performance of our approach with light-weighted training, effectively reducing model hallucinations (no hallucination rate wit a gain from 40% to 56%) and significantly improving the accuracy of model descriptions (BLEURT score increased from 41.2 to 43.3).

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

## A  DETAILED ALGORITHM DESCRIPTION

Three training stages are scheduled in our experiment. We will briefly show this with an algorithm table. Here we take video caption task as example. Where A shows stage 1 while A shows stage 2.

---
**Algorithm 1** Stage 1
---
1: **Dataset**: 1.9M sentences, covering 0.5M summaries (long sentences) and 1.5M captions (short sentences). It should be noted that no testset's sentence are included.
2: **Model**: Text Auto-Encoder, mainly covering BERT and Text Q-Former (5 layers). About 150M trainable parameters, in which 110M are shared with the vision part.
3: **Input**: $text$
4: **repeat**
5:    **Feature Decoupling**: $\mathbf{R}_{text} = \mathbf{QF}_{text}(\text{BERT}(text))$, where $\mathbf{QF}_{text}$ means Text Q-Former
6:    **Loss caculation**: $\mathcal{L}_{text} = \mathbf{LLM}(\mathbf{Proj}(\mathbf{R}_{text}), text).Loss$, where $\mathbf{Proj}$ and $\mathbf{LLM}$ means linear projection and large language model.
7:    **Loss back-propagation and weights updating**
8: **until** convergence
---

---
**Algorithm 2** Stage 2
---
1: **Dataset**: WebVid2M, covering 2M valid (video, text) pairs; CC3M, covering 2.5M valid (image, text) pairs which is considered as static video.
2: **Model**: Text Auto-Encoder, mainly covering BERT and Text Q-Former (5 layers). Video foundation model, mainly covering Image Q-Former, Video Q-Former (5 layers) and ASR BERT. Here ASR BERT are shared by both parts. About 307M trainable parameters, in which only 40M are exclusively belongs to text Auto-Encoder.
3: **Input**: $(text, asr, frames)$, noting we use 'none.' to represent the videos' asr for the pre-training dataset.
4: **repeat**
5:    # A. Text Auto-Encoder pipeline.
6:    **Feature Decoupling**: $\mathbf{R}_{text} = \mathbf{QF}_{text}(\text{BERT}(text))$, where $\mathbf{QF}_{text}$ means Text Q-Former
7:    **Loss caculation**: $\mathcal{L}_{text} = \mathbf{LLM}(\mathbf{Proj}(\mathbf{R}_{text}), text).Loss$, where $\mathbf{Proj}$ and $\mathbf{LLM}$ means linear projection and large language model.
8:
9:    # B. Video pipeline.
10:    **ASR processing**: $\mathbf{R}_{asr} = \text{BERT}(asr)$,
11:    **Video frames processing**: $\mathbf{R}_{vision} = \mathbf{QF}_{img}(\text{ViT}(frames))$, where $\mathbf{QF}_{img}$ means iamge Q-Former in BLIP-2.
12:    **Video feature decoupling**: $\mathbf{R}_{video} = \mathbf{QF}_{video}(\text{Concat}(\mathbf{R}_{vision}, \mathbf{R}_{asr}))$
13:    **Loss caculation**: $\mathcal{L}_{contra}$ is just caculated by the first token from $\mathbf{R}_{video}, \mathbf{R}_{text}$

$$\mathcal{L}_{video} = \mathbf{LLM}(\mathbf{Proj}(\mathbf{R}_{video}), text).Loss$$

$$\mathcal{L}_{align} = \text{MSE}(\mathbf{R}_{video}, \mathbf{R}_{text})$$

14:    # C. Loss combination.
15:    **Total training loss**: $\mathcal{L}_{train} = \mathcal{L}_{video} + \mathcal{L}_{contra} + \lambda_1 \mathcal{L}_{text} + \lambda_2 \mathcal{L}_{align}$
16:    **Loss back-propagation and weights updating**
17: **until** convergence
---

The training stage three is the same with stage A, where the only difference is the dataset.

## B  DEMOS

**Frames**: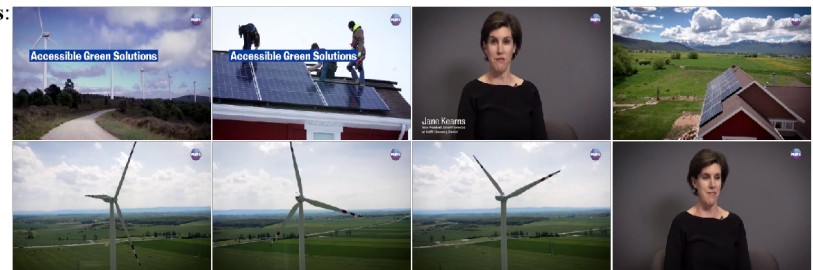

**ASR**：Over the last 10 years, the price of renewable energy has dropped by well over 95%. Renewables are now cheaper than fossil fuels in broad swaths of the world. So I think the biggest surprise in clean tech is that 10 years ago, everybody thought that it was never going to be economic. And what's happened is we've figured out the technologies and the business models that actually make environmental technologies not just better for the planet, but in fact, the most economic solution available out there.

**Video caption**:

**Video-Teller w/o align**: renewable energy is the future, but it's not always the most cost-effective solution.

**Video-Teller** : renewable energy is no longer a cost-prohibitive alternative to fossil fuels.

**Human annotated**: renewable energy's dramatic cost reduction has made it cheaper than fossil fuels, revolutionizing the cleantech industry.

**Video summary**:

**Video-Teller w/o align**: in this video, the speaker highlights the significant decline in the price of renewable energy over the past decade, from around $100 per megawatt-hour in 2010 to less than $10 per megawatt-hour in 2019. the shift from fossil fuels to renewable energy is occurring at a faster pace than many expected, and the speaker attributes this to the technological innovations that have been made available to the market. however, the speaker also points out that the environmental benefits of renewable.

**Video-Teller** : in this video, the speaker discusses the rapid decline in the price of renewable energy, which is now cheaper than fossil fuels in many parts of the world. this is due to a combination of technological advancements and business models that focus on the technological and business aspects of renewable energy rather than just the environmental benefits. the speaker believes that this trend will have a significant impact on the future of the energy industry.

**Human annotated**: in the past decade, renewable energy prices have significantly decreased by over 95%, surpassing the affordability of fossil fuels in many regions worldwide. contrary to earlier beliefs, the most remarkable revelation in cleantech is the economic viability achieved through technological advancements and innovative business models. environmental technologies have emerged as not only superior choices for the planet but also as the most cost-effective solutions available today.

Figure 4: A case shows improvement of hallucination using fine-grained modality alignment.

