# OpenReview forum: "Video-Teller: Enhancing Cross-Modal Generation with Fusion and Decoupling"
_ICLR.cc/2024/Conference — ICLR 2024 Conference Withdrawn Submission_

### Official Review · Reviewer_wY2h · 2023-10-30

**Soundness:** 3 good
**Presentation:** 3 good
**Contribution:** 2 fair
**Rating:** 3
**Confidence:** 4

**Summary:**

This paper proposes a LLM-based video understanding framework, which uses text autoencoder + text Q-former and designs the corresponding alignment loss to guide the overall framework to accurately understand the input video and ASR text.
Experimental results show that the proposed framework achieves comparable performance with fewer pre-training samples and performs well in the corresponding downstream tasks.

**Strengths:**

1.  This paper proposes a video understanding architecture that improves the encoding performance of the video encoder by introducing a text autoencoder and corresponding alignment losses.
2.  The experimental results demonstrate the effectiveness of the proposed framework in low-resource and zero-shot settings.
3. The overall design is concise and feasible, and the paper is easy to follow.

**Weaknesses:**

1. The reviewer has the most concern about its limited contributions. Fusing the video features with its corresponding audio signals is a common technique, as in [B]. Aligning video with text before feeding them into a joint Transformer (LLM in this paper) is also a well-known technique, as shown in [A]. Generally speaking, the paper combines multiple existing techniques and the only significant difference is that the authors use the LLM framework which has been popular since 2023.

> [A] Align and Prompt: Video-and-Language Pre-training with Entity Prompts. CVPR 2022.

> [B] MERLOT Reserve: Multimodal Neural Script Knowledge through Vision and Language and Sound. CVPR 2022.

2. The second concern is about the incomprehensive experiments.

(1) This paper compares with existing methods on the data resources used for pre-training, but does not mention the difference in model parameters.

(2) Some experimental results are confusing and lack of explanations, such as the performance of contrastive learning loss in different datasets in Table 4.

(3) The baseline methods used in the experiments lack some SOTA methods, and their performance advantages are not significant compared to the training cost. At least, BLIP-2 should be compared on these benchmarks as it is the baseline.

(4) One important ablation study is missed, where only ASR text is used without visual input. There might exist a shortcut that most of the knowledge is carried by ASR text instead of well-aligned visual features.

(5) The scalability of the method on larger datasets or larger backbones has not been studied. Moreover, it achieves lower accuracy compared to previous methods without LLMs as shown in Tab.1. Its practicality is questionable.

**Questions:**

1.  Do the advantages in low-resource settings come from the introduction of LLM?
2.  Can video-teller handle long videos?

Minor: there are some typos, such as the "black" in Table 3 and the "21.9" of B@4 in Table 4.

---

### Official Review · Reviewer_2rTV · 2023-10-30

**Soundness:** 2 fair
**Presentation:** 2 fair
**Contribution:** 2 fair
**Rating:** 3
**Confidence:** 4

**Summary:**

This work centers on video captioning and presents a framework in the style of BLIP2 to incorporate a large language model. The authors conducted experiments on two public datasets and demonstrated some improvements.

**Strengths:**

The paper is highly accessible and straightforward.
The large language model (LLM) excels in reasoning and demonstrates its capability in multi-modality learning. This work attempts to incorporate such a configuration.

**Weaknesses:**

The primary concern raised in this work pertains to the limited scope of experiments conducted. While the study does include experiments on the MSR-VTT and Video-CSR datasets, it's worth noting that widely recognized benchmarks like VATEX, MSVD, and YouCOOK2 are conspicuously absent from the evaluation.

Furthermore, some key baseline models are notably missing from the experimental setup. For instance, the absence of benchmark models like SwinBERT, which offers end-to-end Transformers with sparse attention for video captioning, raises questions about the comprehensiveness of the comparative analysis.

In terms of novelty, the paper appears to present a somewhat weaker case. The introduction of Q-Former is not an entirely new concept, and the integration of large language models (LLMs) into video captioning has already been explored by prior works such as GIT2. This raises the question of how this work distinguishes itself from existing research in the field.

**Questions:**

- The ablation study about loss weight is not included.

---

### Official Review · Reviewer_9ESt · 2023-10-31

**Soundness:** 2 fair
**Presentation:** 3 good
**Contribution:** 2 fair
**Rating:** 5
**Confidence:** 4

**Summary:**

A video-text foundation model named Video-Teller, which employs the video-text modality fusion and alignment to solve the cross-modality generation task,  is proposed in this paper. In general, the authors adopt the combination of Q-Former and Large Language Model (LLM) to incorporate visual and text information. Experiments on three different tasks, including video captioning, video summarization, and video retrieval, are conducted to evaluate the effectiveness of the proposed method.

**Strengths:**

- The manuscript is generally easy to follow.
- Comprehensive experiments on diverse tasks are conducted to verify the effectiveness of the proposed method.
- The superiority of the proposed method on the video summarization task in comparison to the state-of-the-art methods is impressive.

**Weaknesses:**

Firstly, my main concern lies in the limited contribution. As stated in the contribution part of this paper, the decrease in the computational cost of the proposed method is attributed to the usage of the cascaded Q-Formers. However, from what I can gather, there seems to be no additional improvement in the design of the existing Q-Former. Furthermore, employing a text auto-encoder with LLM as the decoder to decouple textual features appears to be a commonly used strategy in text-to-video generation tasks. The novelty of this paper is not clearly demonstrated to the readers.

Secondly, the technical details of the proposed method are either vague or skipped. For instance, there is no explanation provided on how the Q-Formers are cascaded to form the video and text encoder. Additionally, the specific type and architecture of the prompt Large Language Model (LLM) adopted in this paper are not clarified.

Thirdly, no experiments are conducted to verify the claimed decrease in computational cost resulting from the proposed method. The only experimental result I could find related to this is the reduced amount of training data required in the proposed method compared to state-of-the-art methods. Please correct me if I have overlooked any relevant content.

Finally, I have some questions regarding the experimental settings in this paper. For each task, the control methods used for comparison with the method proposed in this paper are different and there are also differences in their numbers, as illustrated in Table 1, Table 3, and Table 7. There is a lack of explanations on why these control methods were chosen.

**Questions:**

My questions are raised in the weakness section. please refer to such content and reply to my concerns.

---

### Official Review · Reviewer_5do8 · 2023-11-05

**Soundness:** 3 good
**Presentation:** 2 fair
**Contribution:** 2 fair
**Rating:** 5
**Confidence:** 4

**Summary:**

This study introduces a video-language foundation model that harnesses the potential of multi-modal fusion and intricate modality alignment, aiming to refine the video-to-text generation endeavor. It finds its foundational roots in the image-centric BLIP-2 model, further enriched by the integration of the cascaded Q-Former, a mechanism adept at consolidating information from video frames and ASR transcripts. Empirical evaluations across tasks such as video captioning, summarization, and retrieval provide insightful results, suggesting the relative efficacy of the model when benchmarked against contemporaneous contributions, notably VideoChat and Video-LLaMA.

**Strengths:**

1. The initiative to incorporate LLM in the creation of an extensive video-language foundation is discernibly a forward-thinking approach with significant potential.

2. The model's rigorous empirical assessment across diverse tasks, such as video captioning, summarization, and retrieval, offers a well-rounded perspective. This extensive evaluation underscores the model's versatility, ensuring its capabilities are thoroughly understood and aptly validated for a variety of applications.

3. The integration of the cascaded Q-Former marks a noteworthy advancement. Crafted to fluidly amalgamate information from both video frames and ASR transcripts, this mechanism suggests promising prospects for enhancing the quality of information retrieval and representation within the video-language domain.

**Weaknesses:**

1. A primary point of contention pertains to the element of novelty. At its core, this study appears to be a straightforward adaptation of BLIP2 tailored for the video realm. The inclusion of the Q-former, while intriguing, is not entirely groundbreaking, as it's an integral component of BLIP2. Furthermore, the training objectives, which encompass intermediate contrastive learning and auto-regressive text generation, have been comprehensively explored in both BLIP2 and CoCa.

2. In terms of practical outcomes, there's room for enhancement. Specifically, when observing video summarization, the zero-shot outcomes do not stand out, especially when benchmarked against solutions like VideoChat. Additionally, in the context of video captioning on MSRVTT, the results lag behind the performance metrics set by GIT2.

3. Why not incorporate a comparative analysis with task-specific methodologies? This would provide a clearer perspective on the relative efficacy of the video foundation model, shedding light on whether it indeed offers superior results in specific contexts.

4.  The review did not mention any in-depth error analysis, which is crucial in understanding where the model falls short and how it can be improved.

**Questions:**

1. Figure 1 appears to echo the content of Figure 2, leading to redundancy in visual representation. A more impactful approach might be to leverage Figure 1 to delve deeper into task-specific motivations or insights. This would not only differentiate the content of the two figures but also offer readers a more granular understanding, rather than just presenting an overarching framework overview. Such an enhancement could streamline the content and enrich the narrative of the research.

2. Missing reference like ChatVideo [1].

[1]https://arxiv.org/pdf/2304.14407.pdf